# Diamagnetic mechanism of critical current non-reciprocity in multilayered superconductors

Ananthesh Sundaresh[1], Jukka I. Väyrynen [1], Yuli Lyanda-Geller[1] & Leonid P. Rokhinson [1,2] ✉

The suggestion that non-reciprocal critical current (NRC) may be an intrinsic property of non-centrosymmetric superconductors has generated renewed theoretical and experimental interest motivated by an analogy with the non-reciprocal resistivity due to the magnetochiral effect in uniform materials with broken spatial and time-reversal symmetry. Theoretically it has been understood that terms linear in the Cooper pair momentum do not contribute to NRC, although the role of higher-order terms remains unclear. In this work we show that critical current non-reciprocity is a generic property of multilayered superconductor structures in the presence of magnetic field-generated diamagnetic currents. In the regime of an intermediate coupling between the layers, the Josephson vortices are predicted to form at high fields and currents. Experimentally, we report the observation of NRC in nanowires fabricated from InAs/Al heterostructures. The effect is independent of the crystallographic orientation of the wire, ruling out an intrinsic origin of NRC. Non-monotonic NRC evolution with magnetic field is consistent with the generation of diamagnetic currents and formation of the Josephson vortices. This extrinsic NRC mechanism can be used to design novel devices for superconducting circuits.

Diodes are the most basic elements of semiconductor electronics and development of superconducting diodes can extend the functionality of the superconducting circuitry. A non-reciprocal critical current (NRC) in a multiply-connected superconductors is a well known effect and can be readily observed in, e.g., an asymmetric superconducting rings[1]. An implicit suggestion that NRC may be an intrinsic property of non-centrosymmetric superconductors[2] generated a renewed theoretical and experimental interest motivated by an analogy with the non-reciprocal resistivity due to the magnetochiral effect, which can appear in uniform materials with broken spatial and time-reversal symmetry[3]. However, a direct analogy between corrections to resistivity and superconducting current is misleading because the anisotropy of scattering is caused by the spin-orbit effects, while the proposed origin of nonreciprocity in singlet-pairing superconductors

is a spin-independent Lifshitz invariant[4,5]. It has been demonstrated in the literature[6] that in uniform singlet superconductors in constant magnetic field the Lifshitz invariants can be eliminated by a gauge (Galilean) transformation from both the Ginzburg-Landau (GL) equation and the expression for the supercurrent, so that linear in Cooper pair momentum terms do not lead to nonreciprocity. Phenomenological treatment shows that cubic in the Cooper pair momentum terms can lead to NRC corrections[7,8]. It has been also suggested that the Rashba terms in the electron spectrum contribute to nonreciprocity within the formalism of quasiclassical Eilenberger equations[9]. A symmetry analysis and microscopic calculations of the Cooperon propagator in the presence of the Zeemann effect and a linear or cubic Dresselhaus spin-orbit interactions in the electron spectrum show that the NRC magnitude and sign depend on the

[1]Department of Physics and Astronomy, Purdue University, West Lafayette, IN 47907, USA. [2]Department of Electrical and Computer Engineering, Purdue University, West Lafayette, IN 47907, USA. ✉e-mail: leonid@purdue.edu

crystallographic orientation of the supercurrent flow (Y.L.G., J.I.V., A.S. & L.P.R., manuscript in preparation). Such anisotropy should characterize non-reciprocity in superconductor/semiconductor heterostructures and cubic uniform singlet noncentrosymmetric superconductors. Another suggested mechanism of NRC is the formation of non-uniform currents in superconducting multilayers[10,11].

In this paper we show that NRC naturally arises in the presence of the magnetic-field–generated diamagnetic currents when neighboring layers in multilayer superconductors are strongly coupled. The total current through a multilayer structure is divided between the layers as an inverse ratio of their kinetic inductances. Initially only one of the layers reaches the maximum (critical) current as the total current increases. A further current increase in the superconducting state requires generating a phase difference between the layers, which adds Josephson energy penalty to the total energy, and in the case of strong interlayer coupling (large Josephson energy) the whole system transitions into a normal state. Field-generated diamagnetic currents will either increase or decrease an external current where transition to the normal state occurs, thus leading to the NRC. In the regime of intermediate interlayer coupling strengths, an interlayer phase difference can change by $2\pi$, leading to the formation of Josephson vortices. Experimentally, we report observation of NRC in nanowires fabricated from InAs/Al heterostructures. The observed non-monotonic evolution of NRC with magnetic field is consistent with the formation of Josephson vortices. Our findings show that the extrinsic contribution to NRC is generic to multilayer

superconductors, and may provide a compelling explanation to the NRC observed in Refs. [2] and[12], in the latter work magnetic flux produced by a Co layer generates opposite diamagnetic currents in the adjacent Nb and V layers.

The term "superconducting diode effect" has been used to describe NRC in different systems, including thin superconducting films[13–18] and Josephson junctions[19–27]. In the former experiments the presence of out-of-plane magnetic field and formation of vortices is essential for the observation of NRC, in this case the critical current is determined by the strength and symmetry of the flux pinning potential. In the latter case the critical current in Josephson junctions is determined by the overlap of Andreev states. In this paper we restrict our discussion to the origin of NRC in long nanowires, where critical current is determined by the depairing velocity of Cooper pairs (the Bardeen limit[28]).

## Results

We have studied switching currents $I_{sw}$ defining a transition from superconducting to normal state in nanowires fabricated from Al/InGaAs/InAs/InGaAs heterostructures[29], where patterned Al top layer forms a nanowire and induces superconductivity in a high mobility InAs quantum well via the proximity effect. An AFM micrograph of a typical device is shown in the inset in Fig. 1. A typical current-voltage characteristic exhibits a sharp switching transition limited by the current resolution (< 5 nA for the fastest sweep rates used in our experiments). A histogram of switching currents $I_{sw}^{\pm}$ for positive (+) and negative (-) current sweeps is shown in Fig. 1a for 10,000 sweeps. Field

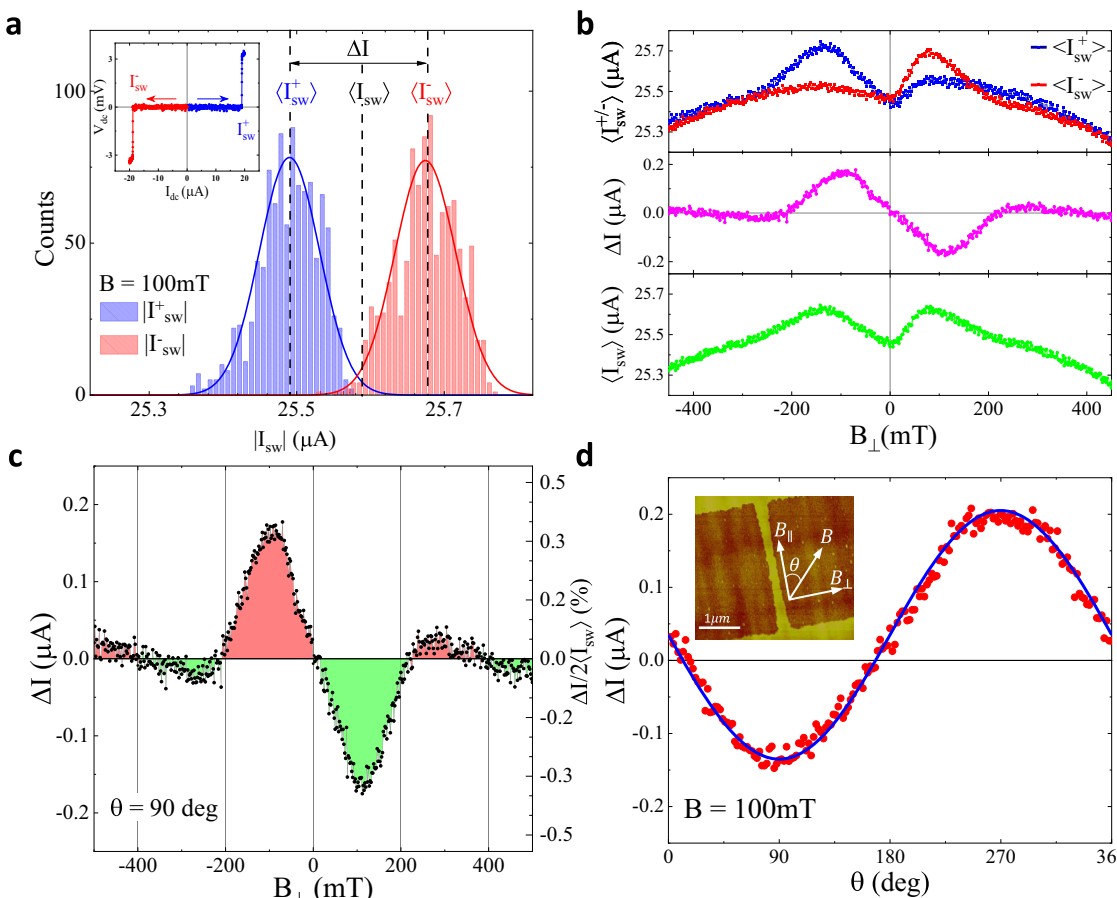

**Fig. 1 | Nonreciprocal critical current in Al/InAs nanowires. a** Histograms of switching currents for 10,000 positive $I_{sw}^{+}$ and negative $I_{sw}^{-}$ current sweeps performed at $T = 30$ mK and $B_{\perp} = 100$ mT. Inset shows a typical current-voltage characteristic. **b** Average switching current for positive $\langle I_{sw}^{+}\rangle$ and negative $\langle I_{sw}^{-}\rangle$ sweeps, non-reciprocal difference $\Delta I = \langle I_{sw}^{+}\rangle - \langle I_{sw}^{-}\rangle$ and an average of all sweeps $\langle I_{sw}\rangle$ is plotted as a function of in-plane magnetic field $B_{\perp}$. In (**c**) enlarged $\Delta I$ data is colored to signify non-monotonic field dependence and multiple sign changes. **d** Dependence of $\Delta I$ on in-plane field orientation is measured at a constant $B = 100$ mT. Blue line is a fit with a sine function. Insert shows an AFM image of a 3μm-long wire connected to wide contacts, yellow areas are Al, in darker areas Al is removed and InAs is exposed.

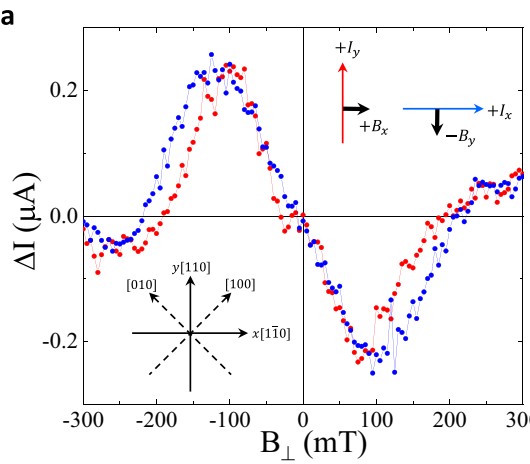

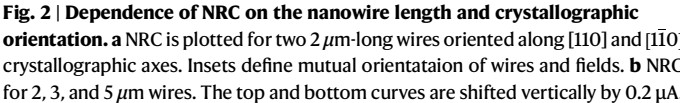

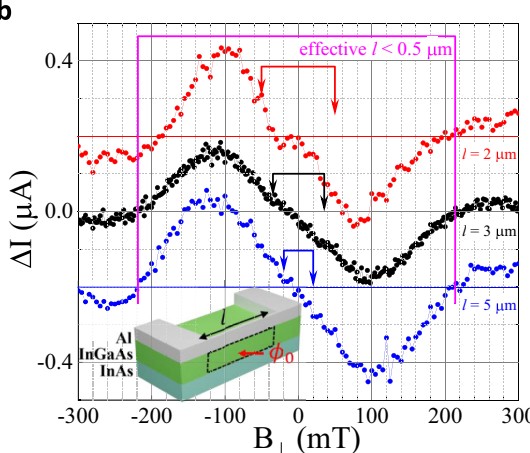

**Fig. 2 | Dependence of NRC on the nanowire length and crystallographic orientation. a** NRC is plotted for two 2 $\mu$m-long wires oriented along [110] and [1$\bar{1}$0] crystallographic axes. Insets define mutual orientataion of wires and fields. **b** NRC for 2, 3, and 5 $\mu$m wires. The top and bottom curves are shifted vertically by 0.2 $\mu$A.

Brackets with arrows indicate a maximum $\Delta B$ needed to insert a flux $\phi_0 = h/2e$ in the area defined by the corresponding wire lengths, as indicated by a dashed loop in the inset. An effective length for the period marked by a magenta bracket is $l = 0.5\mu$m for the same loop.

dependence of average values $\langle I_{sw}^+ \rangle$ and $\langle I_{sw}^- \rangle$ is plotted in Fig. 1b for the in-plane field $B_\perp$ perpendicular to the wire. The $\langle I_{sw}^+ \rangle$ and $\langle I_{sw}^- \rangle$ can be separated into a symmetric $\langle I_{sw} \rangle = (\langle I_{sw}^+ \rangle + \langle I_{sw}^- \rangle)/2$ and asymmetric $\Delta I = \langle I_{sw}^+ \rangle - \langle I_{sw}^- \rangle$ parts, the latter being the non-reciprocal component of the supercurrent. Both $\langle I_{sw} \rangle$ and $\Delta I$ are nonmonotonic functions of magnetic field. As shown in the Supplement, a minima of $\langle I_{sw} \rangle$ at low fields vanishes above 350 mK (0.3 $T_C$), while there is no change in $\Delta I$ at least up to 750 mK (>0.6$T_C$). This difference in energy scales for the appearance of NRC and non-monotonic evolution of $\langle I_{sw} \rangle$ indicates that these are unrelated phenomena, and below we focus on the origin of NRC. Some devices were fabricated with a top gate, which allows electrostatic control of the electron density in the InAs layer not covered by Al; we found that depletion of the 2D electron gas in the exposed InAs results in a slight increase of $\langle I_{sw} \rangle$ but does not affect $\Delta I$. Similar field effect has been observed previously in superconductor nanodevices[30] and was attributed to the presence of quasiparticles[31], a conclusion consistent with the observed gate dependence of the $\langle I_{sw} \rangle$.

Unlike the linear in Cooper pair momentum terms, higher order terms cannot be removed by gauge transformation and it was shown that the presence of terms - $\alpha_3 Q^3 \Delta^2$ cubic in the Cooper pair momentum in an expansion of the Ginsburg-Landau coefficients can generally lead to non-zero $\Delta I$ which is a non-monotonic function of $B$ and can even change sign[7,9] (here $\mathbf{Q} = -i\hbar\nabla - 2e\mathbf{A}$ is a generalized Cooper pair momentum, $\mathbf{A}$ is electromagnetic vector-potential). However, for proximitized InAs layer, a generation of the terms higher order in the Cooper pair momentum in the presence of the Rashba spin-orbit and Zeeman interactions coexists with a similar generation of such terms due to the Dresselhaus spin-orbit interactions. The importance of the Dresselhaus-like terms in the electron spectrum is not limited to proximity structures, and they can play significant role in any noncentrosymmetric material. Investigation of realistic cubic terms in the Cooper pair momentum showed (Y.L.G., J.I.V., A.S. & L.P.R., manuscript in preparation) that nonreciprocity becomes highly anisotropic as a result of Dresselhaus-induced contribution. For comparison with experiments, it is instructive to express the odd in Cooper pair momentum part of the kinetic energy in coordinates rotated by $\pi/2$ with respect to the principal crystallographic axes of InAs, where $\hat{x}\|[1\bar{1}0]$ and $\hat{y}\|[110]$, see insert in Fig. 2a. In these coordinates, the cubic in the Cooper pair momentum kinetic term originating from the cubic Dresselhaus electron spin-orbit interaction reads

$$f_k = |\kappa \left( B_y Q_x^3 + B_x Q_y^3 - Q_x Q_y [B_x Q_y + B_y Q_x] \right) \Delta|^2, \quad (1)$$

where coefficient $\kappa$ contains the Dresselhaus constant $\beta_D$ and other material parameters. The resulting NRC correction to the supercurrent is

$$\Delta I \propto \left( B_y I_x^2 + B_x I_y^2 \right). \quad (2)$$

This correction is independent of the sign of $I$ and is added or subtracted to the $B = 0$ current value depending on the direction of the current flow. Here $B_x$ and $B_y$ enter symmetrically for wires oriented along $x$ and $y$. However, in the configuration with the current $I \| \hat{x}$ and magnetic field $B_y$ and the configuration with $I \| \hat{y}$ and $B_x$, this expression has opposite signs for the same mutual orientation of $I$ and $B$, see inset in Fig. 2a. Thus, the Dresselhaus-induced contribution results in NRC with opposite sign for wires oriented along [1$\bar{1}$0] and [110] crystallographic axis. The cubic (and generally all odd) in Cooper pair momentum terms originating from the Rashba electronic interactions, when added with the Dresselhaus-induced terms, will produce anisotropy in the absolute value of NRC, and, in particular, different values of non-reciprocal asymmetrical component of the current for those two directions. Theoretical investigation of electronic spectra of these systems[32] suggests that in narrow InAs quantum wells cubic Dresselhaus terms are larger than the Rashba terms. The lower limit for the value of the Dresselhaus contribution can be extracted from the total spin-orbit anisotropy (which is defined by the ratio between a linear Rashba, and a linear and cubic Dresselhaus terms in electronic spectrum), which was measured to be 70% in spin-galvanic and circular photogalvanic experiments[33] and > 15% in transport experiments[34,35]. Such anisotropies must result in the corresponding crystallographic anisotropy of the NRC, which is not observed in our experiments, Fig. 2a. Therefore, we conclude that the NRC we observed is not intrinsic. The observed NRC does not depend on the wire length, Fig. 2b, which rules out trivial effects related to the formation of spurious loops due to the presence of wire/contact boundaries.

While recent interest in NRC has been motivated by a possibility of the intrinsic origin of the effect, NRC naturally arises in multiply-connected superconductors. In superconducting loops, the critical current is modulated by an external flux $\phi = BS_{loop}$ piercing the loop. In a loop with asymmetric arms, the current maximum is shifted from $B = 0$, and the sign of the shift depends on the direction of the current as shown in Fig. 3a. A nonreciprocal component of the switching current $\Delta I$ is linear in $B$ in the vicinity of $B = 0$, reaches extrema at $\phi \approx \phi_0/4$, changes sign and oscillates with a period $\Delta\phi = \phi_0$. Thus, an asymmetric loop is the simplest "superconducting diode". There is a clear similarity

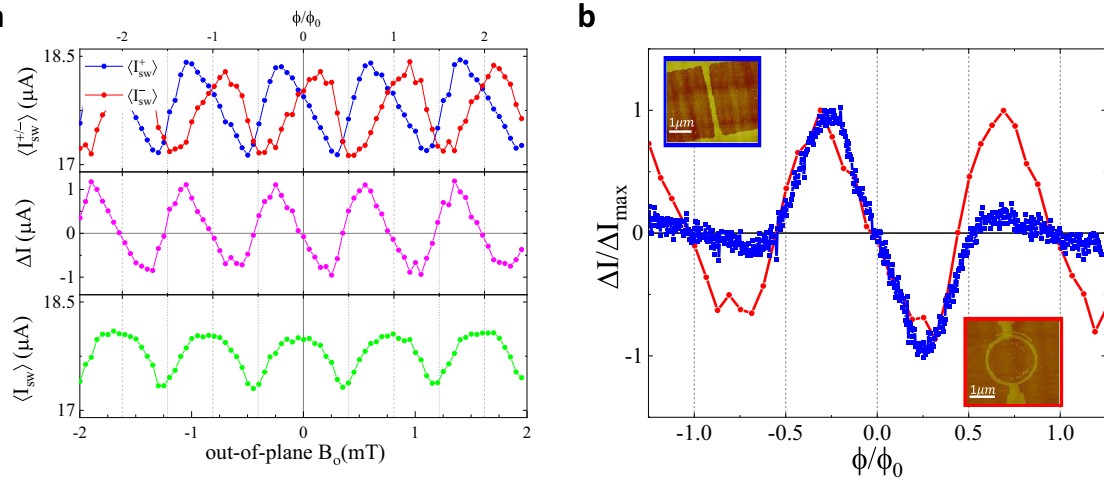

**Fig. 3 | NRC in an asymmetric superconducting loop. a** An average switching current for positive $\langle I_{sw}^{+}\rangle$ and negative $\langle I_{sw}^{-}\rangle$ sweeps, non-reciprocal difference $\Delta I = \langle I_{sw}^{+}\rangle + \langle I_{sw}^{-}\rangle$ and an average of all sweeps $\langle I_{sw}\rangle$ plotted as a function of out-of-plane magnetic field $B_o$ for a loop shown in the insert in (**b**). Note that $\langle I_{sw}\rangle$ is maximal while $\Delta I = 0$ when the flux $\phi = n\phi_0$. In (**b**) $\Delta I$ for the nanowire and the loop are plotted together as a function of a reduced flux $\phi/\phi_0$, where we used $S_{wire} = 0.0052\mu m^2$ for the effective area in the wire and $S_{loop} = 2.59\mu m^2$ in the loop.

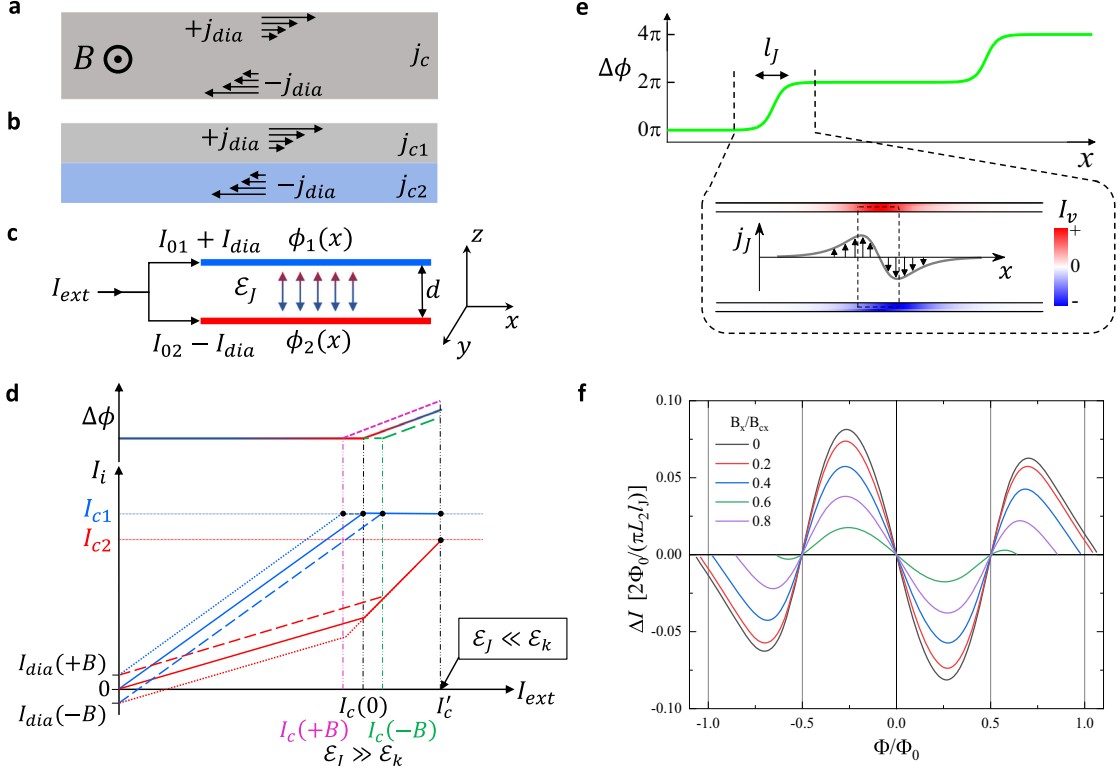

**Fig. 4 | Nonreciprocity of the critical current in the presence of diamagnetic currents.** Diamagnetic currents in (**a**) a uniform superconductor and (**b**) a heterostructure. **c** A two-layer heterostructure is modeled as two zero thickness wires separated by a distance $d$ with coupling described by the Josephson energy $\mathcal{E}_J$. **d** Schematic of current distribution between the wires $I_i = I_{0i} - (-1)^i I_{dia}$, $i = 1, 2$, and the phase difference $\Delta\phi$ as a function of an external current $I_{ext} = I_1 + I_2$ for $B = 0$ (solid lines), $B > 0$ (dotted lines) and $B < 0$ (dashed lines). For weakly coupled wires $\mathcal{E}_J \ll \mathcal{E}_k$, the critical current is field-independent $I_c' = I_{c1} + I_{c2}$, see the text; the critical current is reduced and acquires a linear-in-$B$ correction in a strong coupling regime $\mathcal{E}_J \gg \mathcal{E}_k$ due to the phase locking $\Delta\phi = 0$. **e** In the intermediate coupling regime $\mathcal{E}_J \sim \mathcal{E}_k$ Josephson vortices may form generating a $2\pi$ phase twist, in this case NRC becomes a non-monotonic function of $B$. **f** Calculated NRC $\Delta I$ is plotted as a function of flux $\Phi = S_v B_y$ for several $B_x$, Eq. (S12).

between $\Delta I$ measured in an asymmetric superconducting loop and in an Al/InAs nanowire as emphasized in Fig. 3b, suggesting that non-monotonic NRC in our nanowires may be due to emerging current loops.

External magnetic field generates circular diamagnetic currents in a superconductor, as shown schematically in Fig. 4a,b, and these currents affect the critical current. In homogeneous superconductors the presence of diamagnetic currents will not result in the critical current non-reciprocity, but in a heterogeneous superconductor, in general, their presence will lead to NRC. Qualitatively, the origin of NRC can be understood from a simplified model of a superconductor heterostructure represented as two coupled zero-thickness

superconducting wires separated by a distance $d$, Fig. 4c. The total energy of the two-wire system can be written as a sum of kinetic and Josephson energies,

$$E_{tot} = \int dx[\mathcal{E}_k - \mathcal{E}_J \cos(\Delta\phi)], \qquad (3)$$

where $\mathcal{E}_k = L_1 I_1^2 + L_2 I_2^2$, $\mathcal{E}_J$ is the Josephson coupling, $\Delta\phi = \phi_1(x) - \phi_2(x)$ is the phase difference between superconducting condensates, and $L_i$ are the kinetic inductances per unit length in wires labeled by an index $i = 1, 2$. Supercurrents in each wire $I_i = (2eL_i)^{-1}(\hbar\partial_x\phi_i - 2eA_x)$ should satisfy charge conservation constraint $I_1(x) + I_2(x) = I_{ext}$, where $I_{ext}$ is the applied external current. Detailed solution for this model can be found in the Supplementary Materials, and we outline now the main results. For small external currents ($I_1 < I_{c1}$ and $I_2 < I_{c2}$, where $I_{ci}$ are the critical currents in the wires) it is energetically favorable to keep the phase difference $\Delta\phi$ constant ($\Delta\phi = 0$ for $\mathcal{E}_J > 0$). Then, the currents can be expressed as $I_1 = I_{01} + I_{dia}$ and $I_2 = I_{02} - I_{dia}$, where $I_{01}, I_{02} \propto I_{ext}$ with $I_{01}/I_{02} = L_2/L_1 = \eta^{-1}$ and $I_{dia} = B_y d/(L_1 + L_2)$. Dependence of $I_1$ and $I_2$ on $I_{ext}$ for $B_y > 0$, $B_y = 0$ and $B_y < 0$ is plotted schematically in Fig. 4d. As $I_{ext}$ increases and one of the currents ($I_1$ in our example) reaches the critical value $I_{c1}$, further external current increase requires an increase of $|\Delta\phi|$ because the excess current has to flow through the remaining superconducting wire with the current $I_2$. In the case of weak interwire coupling, $\mathcal{E}_J \ll \mathcal{E}_k$, deviation of $\Delta\phi$ from zero does not lead to a significant energy penalty and the critical current of the whole system $I_c' = I_{c1} + I_{c2}$ does not depend on the magnetic field direction. In the opposite regime of strong coupling, $\mathcal{E}_J \gg \mathcal{E}_k$, the energy cost associated with the formation of Josephson currents (the last term in Eq. (3)) is prohibitively high and the whole system transitions to a normal state at $I_{ext} \approx (1 + \eta)(I_{c1} - I_{dia})$, resulting in $\Delta I = -2(\eta + 1)I_{dia}(B_y)$ (this equation is correct for $\beta > \eta + (\eta + 1)I_{dia}/I_{c1}$, where $\beta = I_{c2}/I_{c1}$, NRC for other scenarios is listed in the Supplementary Materials). Thus a superconducting diode effect is a generic property of coupled multilayer superconductors.

As $B_y$ and diamagnetic currents increase, the phase locking condition $\Delta\phi = 0$ along the length of the wires leads to a significant increase of $\mathcal{E}_k$. At a critical field $B_c = (3/\pi^2)\Phi_0/(l_J d)$ it becomes energetically favorable to reduce the overall energy by twisting the phase difference by $2\pi$ locally forming a Josephson vortex ($l_J \approx \Phi_0/(2\pi\sqrt{2\mathcal{E}_J L_2})$ and $\Phi_0 = h/2e$ is the flux quantum). Evolution of the phase difference $\Delta\phi(x) = 4\arctan[\exp(x/l_J)]$, vortex-induced currents in the wires $I_v(x)$, and interwire Josephson current density $j_J(x)$ across a vortex are shown schematically in Fig. 4e. The maximum of $I_v(x)$ at the center of the vortex determines the Josephson vortex contribution to NRC. In the absence of quantum fluctuations formation of a vortex is accompanied by an abrupt re-distribution of currents between the wires, which results in a sawtooth NRC dependence on the magnetic field. Generation of multiple Josephson vortices does not modify NRC compared to a single vortex case unless the vortices significantly overlap so that the maximum of $I_v(x)$ exceeds its single-vortex value.

In Fig. 4f we plot $\Delta I(B_y)$ for several $B_x$ using Eq. (S12) in the Supplemental Material. A gradual change of $\Delta I$ near $\Phi_0/2$ is due to quantum fluctuations of the winding number due to strong coupling of the vortex to current-carrying wires. This smearing is similar to the gradual change of a critical current in a ring connected to superconducting leads (Fig. 3), as compared to an abrupt reversal of persistent currents at $\Phi_0/2$ in isolated rings[36]. The period of oscillations of $\Delta I$ corresponds to the flux threading an effective vortex area $S_v = (\pi^2/3)l_J d = l_v d$. The period $\Delta B_\perp = 400$ mT translates into the length $l_v \approx 500$nm, where $\Delta\phi$ substantially deviates from zero. We estimate $l_v < \xi_{InAs} = \sqrt{\xi_{InAs}^0 l_{InAs}^m} \approx 750$ nm and expect proximity-induced superconductivity in InAs to be preserved in the presence of a vortex. Here we use $\xi_{InAs}^0 = \hbar v_F/\pi\Delta^* \approx 1.8\,\mu$m, induced gap in InAs

$\Delta^* \approx \Delta = 1.796 k_B T_c = 230\mu eV$ (induced gap is close to the gap of Al in these heterostructures[37]), and the mean free path in uncapped InAs 2D gas $l_{InAs}^m \approx 300$ nm.

Finally, we use the two-wire model to estimate the temperature and in-plane field $B_\parallel \parallel \hat{x}$ dependences of NRC assuming that both parameters affect the Cooper pair density $n_2$ in InAs. In the vicinity of $B_\perp = 0$ the amplitude of $\Delta I \propto L_2^{-1} \propto n_2$ and is expected to decrease with an increase of $T$ or $B_\parallel$. The critical field $B_c \propto \sqrt{\mathcal{E}_J L_2}$ depends on $\mathcal{E}_J \propto n_2$, and the period of oscillations is expected to be $T$- and $B_\parallel$-independent, Fig. 4d. Josephson coupling $\mathcal{E}_J$ is exponentially sensitive to the thickness of the InGaAs spacer and we expect slight variations of the period $\Delta B_\perp$ between the samples. These qualitative estimates are consistent with experimental observations, see Figs. S2 and S3 in the Supplemental Material.

## Methods

### Materials
The wafer was grown using Molecular Beam Epitaxy (MBE) on an InP substrate. The heterostructure consists of 1 μm graded $In_xAl_{1-x}As$ insulating buffer followed by a $In_{0.75}Ga_{0.25}As(4nm)/InAs(7nm)/In_{0.75}Ga_{0.25}As(10nm)$ multilayer structure capped in-situ with 7nm of Al. The two-dimensional electron gas has a peak mobility of 28000 cm²/Vs at a density $8 \times 10^{11}$ cm⁻².

### Sample fabrication
The nanowires were fabricated using standard electron beam lithography. The mesas were defined by first removing the top Al layer with Al etchant Transene D and then a deep wet etching using $H_3PO_4:H_2O_2:H_2O:C_6H_8O_7$ (1ml:8ml:85ml:2g). Nanowires are defined in the second step of lithography by patterning the Al layer. Some devices have a top electrostatic gate, in these devices a 20 nm $HfO_2$ is grown by atomic layer deposition followed by a deposition of a Ti/Au (10/100 nm) gate.

### Measurements
Current-voltage sweeps were performed using a homemade high-speed high resolution DAC/ADC (digital-to-analog and analog-to-digital converter) setup. The sweeps were automatically interrupted at the superconductor-normal transition ($I_{sw}$) in order to minimize device heating. Current sweep rate and delay between sweeps have been optimized to obtain <5 nA current resolution and to keep device temperature <50 mK at the base temperature of the fridge. The data has been corrected for an instrumental cooldown-dependent constant current offset (generated in the circuit by uncompensated voltages in the system and limited by a 100 kΩ current source resistor) to insure that $\Delta I = 0$ at $B = 0$.

## Data availability
All the data that support the findings of this study are available from the corresponding author on reasonable request. Source data are provided with this paper.

## Code availability
The code used for the numerical simulation has been provided with this paper as Supplementary Data 1.

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

## Acknowledgements

The Al/InAs heterostructures were provided by Michael Manfra Group at Purdue University. The authors thank Lev Ioffe and Igor Aleiner for stimulating discussions. Experimental part was initially supported by the U.S. Department of Energy, Office of Basic Energy Sciences, Division of Materials Sciences and Engineering under Award DE-SC0008630, the work was completed with the support by NSF award DMR-DMR-2005092 (A.S. and L.P.R.). Theoretical work is supported by the U.S. Department of Energy, Office of Basic Energy Sciences, Division of Materials Sciences and Engineering under Award DE-SC0010544 (Y.L-G) and the Office of the Under Secretary of Defense for Research and Engineering under award number FA9550-22-1-0354 (J.I.V.).

## Author contributions

L.P.R. conceived, A.S. performed experiments, J.I.V. and Y.L.G. developed the theory. All authors participated in writing the manuscript.

## Competing interests

The authors declare no competing interests.
