## [Peer Review File · Nature Communications]

REVIEWER COMMENTS

Reviewer #1 (Remarks to the Author):

The paper by Sundaresh et al. describes (experimentally and theoretically) an orbital mechanism for the origin of supercurrent rectification in plain superconductor-proximitized semiconductor heterostructures. This mechanism is based on the nucleation of a Josephson vortex when an in-plane field perpendicular to the current is applied, which produces counterpropagating currents in the two coupled superconducting layers under study.

Owing to the attention attracted by the topic of supercurrent rectification, I think the manuscript is interesting and could potentially deserve publication in Nature Communication. However, I see serious problems with its relation with the literature, which in my opinion require critical changes in the manuscript. My questions and objections are listed in the following.

1. The Authors claim at end of the abstract that the manuscript provides a compelling explanation to previously published results. Which ones?

To the best of my knowledge, the situation described by the authors does not match any of the systems on which the supercurrent diode effect was reported in the literature. The Ref. 1 is a superlattice where the layer spacing is much less than the coherence length ξ , and the interlayer coupling is very strong. Going through the manuscript reference list, Ref. 23 is the one with the most similar material, but in that paper the authors study Josephson junctions (in the sense that along the x direction there is an alternation of S and N regions, the rectification is described by completely different physics and probed experimentally also by other means, i.e., Josephson inductance asymmetry). Besides that, it does not seem to me that experiments in the literature have studied the system studied by the authors (namely a bilayer of different SCs with mutual Josephson coupling neither too strong nor too weak), so the aforementioned statement of the authors at the end of the abstract is not clear to me.

2. Things become more critical in the paragraph starting with "In this paper...". The authors write that "critical current in thin films in out-of-plane magnetic fields [14-16] or in superconductors in proximity to a ferromagnet [17-19] is limited by flux pinning and NRS can be attributed to ratchet effect from an asymmetric pinning potential." Here the authors attribute the NRS in the six references 14-19 to ratchet effect. I read the papers and none of the six papers refers to this effect. (Ref. 17 is not even about supercurrent rectification: it is an ordinary resistive rectification due to spin-filtering effect of an NS junction, no supercurrent is involved here).

Citation of papers in the literature is a delicate operation and must be done with the greatest care. (Of course, it is still possible and legitimate that the authors want to claim that, in contrast to what is written in these papers, the NRS effect there is indeed due to ratchet effect: but then the authors have to carefully demonstrate this statement, disproving all the evidences provided by each of these papers. I do not think, however, this is the scope of the authors in the present manuscript.)

Continuing on the same paragraph the authors write:

"In Josephson junctions the interplay between magnetic field and spin-orbit interactions results in a non-reciprocal Doppler shift of Cooper pair momentum [22] and NRS due to phase rigidity in contacts [23-25], see also Refs. [26, 27]."

Similarly to what discussed above, I do not think the References are cited in an appropriate fashion. In Ref 23 the nonreciprocal supercurrent comes from the physics of Andreev bound states in the SNS junction, and the contacts play no significant role. In 24 the description is more phenomenological, but still the NRS comes from the Cooper pair momentum in the nominally N region. Finally, Refs. 26 and 27 look to me misplaced in this particular point of the paper, since I do not see a relation with the phase rigidity of the contacts.

A final remark to this paragraph. Usually the paragraph starting with "In this work..." should present in a concise fashion the results of the paper itself. Review of the literature is more appropriate in the introductory paragraphs.

3. A recent preprint by the Moodera's group (Hou et al. arXiv:2205.09276) found that vortex physics might be responsible of a spurious (or extrinsic) rectification effect. I think the authors

must explain how they made sure that the in-plane field had no out-of-plane components (for instance, is it there a field compensation system?). Also, the authors should motivate why in their opinion Abrikosov vortices, produced by residual out-of-plane field component, play no role.

4. The authors write that $\langle I_c \rangle$ (both + and -) have a $\cos(2\theta)$ dependence. First, to be more precise, this should be a $\cos(2\theta) + \text{constant}$, where the constant is larger than the amplitude of the cosine (as opposed to ΔI which is indeed proportional to $\sin(\theta)$ [see Fig.S1 c and d]). Apart from this detail, it is not clear to me the reason of this $\cos(2\theta)$ dependence, in particular the factor 2 multiplying θ . Importantly, it is not clear if this dependence would support or not the model the authors present for the effect.

5. Fig.4c is not clear, and the caption does not help to clarify. What does exactly the color scale indicate? Δj_x or Δj_y ? (and what is the exact definition of it?). What does the green curve in the middle indicate?

6. In the Supp. Mat. the authors describe their model. I have questions about that (see below), but I want first to stress that this model is at the basis of the understanding of the entire paper, therefore I would move it at least to the "Methods" section (where there should be enough space for it).

7. A central statement of this manuscript is the following:

"It has been already demonstrated in the literature [2] that in uniform superconductors Lifshitz invariants can be eliminated by a gauge transformation from both the Ginzburg-Landau (GL) equation and the expression for the supercurrent. Thus, they cannot lead to non-reciprocal contributions to the critical current or a kinetic inductance."

To my knowledge, this statement is strictly true only for infinite systems: for finite systems gauging out the Lifshitz term modifies the boundary conditions for the order parameter Ψ , therefore the system is critically affected by that. Since any practical system is finite, I would like the authors to comment or clarify this point if they want to make statements about the consequences of Lifshitz invariants.

8. At the end of the main text part, there are several estimates of quantities, which depends on experimental parameters, but it is not clear how they were determined. To be more explicit, I would like to know how ΔI in InAs (this is actually the proximity induced ΔI , usually indicated as ΔI^* - see e.g. Phys. Rev. Appl. 7, 034029 (2017), Kjaergaard et al.). Also, I would like to know how the mean free path l is determined. In the system there are actually two mean free paths, that for the epitaxial Al and that for InAs. But, experimentally, I would guess it is not obvious how to determine them separately.

9. The authors write: "Such mechanism was proposed to explain Josephson current non-reciprocity, in this case $\Delta I \propto \sin(2ql)$, where q is a field-dependent momentum shift and l is the length of the junction[24]. We studied length-dependence of ΔI for wires with $l = 2, 3$ and $5 \mu\text{m}$, Fig. 2(b). The three curves are almost identical, ruling out Doppler shift as an origin of NRS in our devices."

The finite momentum q (producing a so called helical superconducting state) is one of the main mechanisms considered by most theoretical works to explain the SC diode effect in plain films (not Josephson junction). In lateral SNS Josephson junctions the mechanism is slightly different since there is no order parameter in the N region. The supercurrent can be seen as carried by Andreev bound states. Ref 24 uses a phenomenological approach to describe the diode effect in SNS junctions. In Ref 24 the NRS is due to the parameter δ , which is the difference in phase offset (ϕ_0) between the fundamental and the second harmonic in the coupling terms (see Eq. 1-2 in Ref 24). At a microscopic level this corresponds to the difference in the anomalous phase shift between the fundamental and the second harmonic in the current phase relation.

The association made in Ref 24 between δ and ql might be justified on the basis of Ref. 22. But the fact that the NRS depends on the $\sin(2ql)$ is derived in Ref.24 considering a Josephson junction. This allows authors of Ref. 24 to expand the free energy in coupling terms containing the gaps of the two leads. I do not see how to extend this derivation to plain films, therefore the statement " $\Delta I \propto \sin(2ql)$ " in plain films cannot be considered established in the literature.

Thus, the authors should either demonstrate this statement or show other (than [24]) references demonstrating the statement for plain films.

10. In the Supplement the authors write:

"The critical current through the ring is determined by the condition that at large enough I_{ext} , one of the wires (or arms of the ring) turns normal. (Experiment indicates that the switching happens in Al, i.e., wire 1, see below.)"

And then later...

"The measurement shows also that the switching current does not differ much from its $B_{perp}=0$ value (see Fig.S2c), which indicates that the critical current is determined by wire 1, as we assumed in Eq. (S9)".

This latter statement is supposed to demonstrate the former assumption, stated just before Eq. S9, that it is Al that switches first. What it is not clear to me is why the proximitized InAs should not switch normal first (the system would still be superconducting since superconducting Al would shunt it).

List of minor remarks:

- a) Many references have been published, yet they are still indicated with their preprint arxiv number. Please update the manuscript citing references in their final published form.
- b) Many Latex references are broken, both in the main text and in the Supp. Mat., so that reference to some Figures and Equations are missing (?? sign appears). This makes reading difficult. Please fix already at this stage.
- c) The symbol " l " is used both as label for the mean free path and for the total length in $\sin(2ql)$.

Reviewer #2 (Remarks to the Author):

The authors reported on non reciprocal supercurrent transport on Al/InAs nanowire heterostructures. The possibility to achieve supercurrent diode effects, and the study of its origin in several setups, have triggered great interests in the recent literature. Definitely this is a timely topic with a wide interest both for fundamental and more applicative reasons.

The presented observations, and their explanations with an effective two wire models, are of very good quality and scientifically sounds.

However, I have different comments/concerns that the authors should consider before a final assessment can be made:

1) In the abstract and in the introduction it is stressed that in homogeneous superconductor Rashba contributions do not play a role for NRS. However, this is not true in general as it is also mentioned by the authors itself, for instance regarding so-called Josephson diodes (e.g. in the Nat. Nanotech. paper by Baudgardner et al. the dominant role is believed to be due to Rashba contributions). These considerations should be amended and clarified. Furthermore, it is stated that the presented model can explain even older observations, but it is not clear how- and where it is discussed in the paper this point. In addition, many recent works on the present topic are not mentioned. In summary, I definitely suggest a revision of both abstract and introduction in order to clarify both the main results of the present manuscript and to avoid misunderstanding/misconceptions to non expert readers.

2) There is some confusion in the orientation of the magnetic field used in the various measurements and the symbols used in the text. Since this is a relevant point, this should be clarified with care.

3) In Fig.1 the asymmetric switching current contributions does not vanish for $B=0$, why?

4) Have the authors measured also the retrapping current? Does this quantity share analog behaviours?

5) It is mentioned that a small gate modulation is present by acting with a top gate on a region

where Al has been removed and this influence supercurrent amplitude. The authors stated that this is consistent with previous measurements on superconducting film and the presence of quasiparticles. I was wondering if this can be also explained by a simpler Jo-fet like contributions, where the gate simply tune the electron density on a pocket/normal region?

6) It is stated that the symmetric and asymmetric contributions have different temperature dependence with different origins. This is also showed in the supplement, but it is not fully justified. Could the authors provide more quantitative analysis and comments regarding the different origin of the se behaviours?

7) It is often introduced an efficiency parameter in order to estimate rectification, e.g. $\eta = \frac{\Delta I_{sw}}{2 \langle I_{sw} \rangle}$. Could the authors report a quantitative analysis of this quantity?

8) It is not clear to me the figure where different nanowire lengths are studied (to rule out Doppler shift like contributions). Indeed, the investigated configuration is not the same as the ones for JDE. I suggest at least to move this discussion on the supplement, or to remove it .

9) The reported measures are explained in terms of a two wire models, pointing at an extrinsic effect for the observed NRS as in a multiply connected geometry (like in asymmetric squid geometry). However I have found in the SM that the distance d is 10 nm while I cannot find this value in the fabrication methods (it seems to me that there is a factor of two or so); in the supplement it is also stated that the first nw that become normal is the Al nw, called 1-th. However Al is also responsible for proximity effect of the InAs nw: it is somehow counterintuitive that the first superconducting to normal transition takes place in the Al nw; in addition, I'm not sure regarding the conclusions that a reader can infer from this point, and in particular how this can be related to previous multilayer superconductor experiments and NRS. Overall, clear conclusions are missing.

10) (minor comments) all figure labels should be enlarged (especially in the inset); there is some typos in the supplement, e.g. the definition of ΔI_{sw} is not consistent with the one in the main text (wrong sign)

Reviewer #3 (Remarks to the Author):

The manuscript by Ananthesh Sundaresh and co-workers: "Supercurrent non-reciprocity and vortex formation in superconductor heterostructures" investigates the supercurrent diode effect in nanowires fabricated from InAs/Al heterostructures. A clear effect is visible and depends mainly on the in plane magnetic field applied orthogonal to the nanowire. Differing from similar experimental they exclude the role of Rashba and Dresselhaus SO interaction in the asymmetry of the critical current and they develop a model based on the interference between two dissimilar superconductors similarly to an asymmetric SQUID.

I found this piece of work very interesting with a very accurate data collection and analysis. On the other hand, I found the theoretical explanation very leaky with some important consideration on Lifshits invariants that cannot be apply to their experiment. In particular:

1) I found these two sentences wrong: "Theoretically it has been understood that linear-in-momentum energy terms, such as Rashba spin-orbit interaction or, more generally, any aymmetry-allowed Lifshits invariants[2], do not contribute to the supercurrent," and "It has been already demonstrated in the literature [2] that in uniform superconductors Lifshits invariants can be eliminated by a gauge transformation from both the Ginzburg-Landau (GL) equation and the expression for the supercurrent. Thus, they cannot lead to non-reciprocal contributions to the critical current or a kinetic inductance"

It is Known that linear-in-momentum Rashba SO interactions in a BCS superconductor CAN lead to non-reciprocal contribution in the critical current (see for example ref [3] of the manuscript and ref 10.1103/PhysRevLett.128.177001 for a more complete and general analysis). The ref [2] the authors use to exclude linear-in-momentum Rashba SO interaction is only limited to the GL

approximation that is valid only for temperature close to T_c , a temperature range very different from their experiment.

2) Sign changes of the diode effect. It's not true that sign changes are not possible for diode effect based on Rashba SO interaction as mentioned for example in this sentence: "Non-monotonic field dependence and multiple sign changes of ΔI at high fields experimentally further exclude these mechanisms, since otherwise an intrinsic property of the material (vector c) would require to change sign as a function of B ." As shown in ref [3] of the manuscript and ref 10.1103/PhysRevLett.128.177001 a sign change is a peculiar characteristic of SO based diode effects.

3) Model based on the formation of vortices not very convincing. I found the model based on the interference between two superconductors not very convincing in few points:

a. Periodicity of the oscillations: in figure 1c I see two nodes in the ΔI , one at 200 mT and one at 400 mT, is this behavior confirmed in all the other samples? For example, in fig2 a (blue line) I see a node at -220mT and one at 300 mT and in fig S4 b the nodes are at -220 mT and 320 mT. If the origin of these nodes come from interference effects like squids the periodicity should be respected.

b. The model predicts supercurrent diode effect generated by one vortex of 0.5 μm size in the middle of the wire, while the generation of multiple vortices at large fields is excluded. How is it possible? What would be the impact of multiple vortices?

c. Knowing that also SO interaction linear-in-momentum can generate similar features (see point 2) in the same field range (if you consider the large g factor of InAs), how this hypothesis can be excluded? If confirmed this would make the experimental finding even more interesting.

4) Control device. In figure S6 the authors show the measurement on a control device of pure Al claiming that diode effect is not present. But, in panel b a sizable ΔI of -0.4 μA is visible and is larger than the maximal ΔI observed in the devices $\sim 0.2 \mu\text{A}$. How is it possible? In the control case there is no field dependence but the diode effect is larger or comparable with original devices.

Concluding I found the obtained experimental results very novel and important for the emerging field of superconducting spintronics and hybrid topological superconductivity. The data analysis is correct but the interpretation and conclusion are debatable as explained in the previous points. This prohibits the publication in the current state and requires a revision. If the Author can clarify these four points this work will deserve the publication in Nat Communications, once the theoretical part and interpretation is properly amended.

RESPONSE TO REVIEWERS' COMMENTS

From referees' questions and comments we understood that we did not make our main message exquisitely clear to a reader. The central point of the paper is that in coupled multi-layer superconductors non-reciprocity of the critical current naturally arises due to the presence of diamagnetic currents generated by an external magnetic field. The physical origin for this effect is phase locking between layers. The total current through a multi-layer is divided between the layers as inverse ratio of kinetic inductances, and one of the layers reaches its critical current first as the total current increases. Further current increase within the superconducting state requires changing of a phase difference between the layers. In the case of a strong coupling (phase locking) the whole system must transition into a normal state. In a two-layer system diamagnetic current is added to the current in one layer and subtracted from the current of another layer. If diamagnetic current is a small addition to the total current and, e.g., the total current in layer 1 reaches the critical value independent of the magnetic field direction, the presence of diamagnetic currents results in non-reciprocity. Furthermore, in an intermediate coupling regime it becomes energetically favorable to twist phase by 2π in some regions, forming Josephson vortices. Formation of the Josephson vortices at high fields is an interesting prediction of the model which explains non-monotonic evolution of non-reciprocity with magnetic field observed in our experiments. We emphasize, though, that non-reciprocity is a generic phenomena in multi-layer superconductors irrespective of vortex formation. **In order to make this point more transparent, we changed the title, redesigned Fig. 4 and made substantial revisions to the manuscript.**

In the following we address reviewers' questions and comments.

Reply to Reviewer 1

1. The Authors claim at end of the abstract that the manuscript provides a compelling explanation to previously published results. Which ones? To the best of my knowledge, the situation described by the authors does not match any of the systems on which the supercurrent diode effect was reported in the literature. The Ref. 1 is a superlattice where the layer spacing is much less than the coherence length ξ , and the interlayer coupling is very strong. Going through the manuscript reference list, Ref. 23 is the one with the most similar material, but in that paper the authors study Josephson junctions (in the sense that along the x direction there is an alternation of S and N regions, the rectification is described by completely different physics and probed experimentally also by other means, i.e., Josephson inductance asymmetry). Besides that, it does not seem to me that experiments in the literature have studied the system studied by the authors (namely a bilayer of different SCs with mutual Josephson coupling neither too strong nor too weak), so the aforementioned statement of the authors at the end of the abstract is not clear to me.

The main point of our paper is that generation of diamagnetic currents by magnetic field leads to critical current non-reciprocity in multi-layered superconductors. An important condition is phase locking, which is expected in coupled multi-layer structures with layer spacing $< \xi$. This generic mechanism should be present in the multi-layer [Nb/V/Ti] structure studied in Ref [1]. In Ref [23] there is only one layer in the region where critical current is reached (proximatized InAs) and the presence of diamagnetic currents does not lead to critical current non-reciprocity. In the revised manuscript we removed the ambiguous phrase about relevance to previous experiments from the abstract. We now mention that the extrinsic contribution to NRC is generic to multilayer superconductors.

2. Things become more critical in the paragraph starting with "In this paper...". The authors write that "critical current in thin films in out-of-plane magnetic fields [14-16] or in superconductors in proximity to a ferromagnet [17-19] is limited by flux pinning and NRS can be attributed to ratchet effect from an asymmetric pinning potential." Here the authors attribute the NRS in the six references 14-19 to ratchet effect. I read the papers and none of the six papers refers to this effect. (Ref. 17 is not even about supercurrent rectification: it is an ordinary resistive rectification due to spin-filtering effect of an NS junction, no supercurrent is involved here). Citation of papers in the literature is a delicate operation and must be done with the greatest care. (Of course, it is still possible and legitimate that the authors want to claim that, in contrast to what is written in these papers, the NRS effect there is indeed due to ratchet effect: but then the authors have to

carefully demonstrate this statement, disproving all the evidences provided by each of these papers. I do not think, however, this is the scope of the authors in the present manuscript.)

Continuing on the same paragraph the authors write: "In Josephson junctions the interplay between magnetic field and spin-orbit interactions results in a non-reciprocal Doppler shift of Cooper pair momentum [22] and NRS due to phase rigidity in contacts[23–25], see also Refs. [26, 27]." Similarly to what discussed above, I do not think the References are cited in an appropriate fashion. In Ref 23 the nonreciprocal supercurrent comes from the physics of Andreev bound states in the SNS junction, and the contacts play no significant role. In 24 the description is more phenomenological, but still the NRS comes from the Cooper pair momentum in the nominally N region. Finally, Refs. 26 and 27 look to me misplaced in this particular point of the paper, since I do not see a relation with the phase rigidity of the contacts.

A final remark to this paragraph. Usually the paragraph starting with "In this work..." should present in a concise fashion the results of the paper itself. Review of the literature is more appropriate in the introductory paragraphs.

We agree with the referee that detailed analysis of other systems where superconducting diode effect has been observed is outside the scope of this paper and removed it from the revised manuscript. We reviewed citations and made sure that the prior work is appropriately acknowledged.

3. A recent preprint by the Moodera's group (Hou et al. arXiv:2205.09276) found that vortex physics might be responsible of a spurious (or extrinsic) rectification effect. I think the authors must explain how they made sure that the in-plane field had no out-of-plane components (for instance, is it there a field compensation system?). Also, the authors should motivate why in their opinion Abrikosov vortices, produced by residual out-of-plane field component, play no role.

It is important to make a distinction between the Josephson vortices (where the phase difference between two superconducting regions is twisted but the order parameter is not suppressed) and the Abrikosov vortices with non-superconducting core. In Moodera's work, the Abrikosov vortices are formed. It has been shown in earlier papers on thin films that asymmetry in the pinning potential of the Abrikosov vortices can lead to the critical current non-reciprocity, see e.g. [Nat Commun 12, 2703 (2021)].

The Abrikosov vortices can enter our nanowires at a high out-of-plane fields $B_z > 15mT$, and their entrance results in an abrupt decrease of critical current I_c . We use a vector magnet to apply a magnetic field and align the in-plane field within ~ 0.1 degree, which translates into $B_z < 2$ mT for $B = 1$ T. We added a section "In-plane field alignment" to the Supplementary Material, in which the alignment procedure is described.

4. The authors write that (both + and -) have a $\cos(2\theta)$ dependence. First, to be more precise, this should be a $\cos(2\theta) + \text{constant}$, where the constant is larger than the amplitude of the cosine (as opposed to ΔI which is indeed proportional to $\sin(\theta)$ [see Fig.S1 c and d]. Apart from this detail, it is not clear to me the reason of this $\cos(2\theta)$ dependence, in particular the factor 2 multiplying theta. Importantly, it is not clear if this dependence would support or not the model the authors present for the effect.

This is an experimental observation and we do not have a good understanding of the origin of this dependence. As we mention in the text an increase of I_c with magnetic field has been observed before in single layer nanowires. 2θ dependence indicates that I_c enhancement does not depend on the field direction, which may be consistent with polarization of residual impurities with non-zero spin. However, we do not expect any impurities to exceed a few ppm level in these ultraclean heterostructures. Our model does not predict this effect. We changed the caption to $\cos(2\theta) + \text{constant}$ as suggested by the reviewer.

5. Fig.4c is not clear, and the caption does not help to clarify. What does exactly the color scale indicate? Δj_x or Δj_y ? (and what is the exact definition of it?). What does the green curve in the middle indicate?

Fig. 4 in the main text has been redesigned to better explain the model.

6. In the Supp. Mat. the authors describe their model. I have questions about that (see below), but I want first to stress that this model is at the basis of the understanding of the entire paper, therefore I would move it at least to the "Methods" section (where there should be enough space for it).

We added a qualitative description of the model into introduction and details in the main text (a paragraph "External magnetic field generates circular diamagnetic currents in a superconductor, as shown schematically

in Fig. 4(a,b)...”.

7. A central statement of this manuscript is the following: ”It has been already demonstrated in the literature [2] that in uniform superconductors Lifshitz invariants can be eliminated by a gauge transformation from both the Ginzburg-Landau (GL) equation and the expression for the supercurrent. Thus, they cannot lead to non-reciprocal contributions to the critical current or a kinetic inductance.” To my knowledge, this statement is strictly true only for infinite systems: for finite systems gauging out the Lifshitz term modifies the boundary conditions for the order parameter Ψ , therefore the system is critically affected by that. Since any practical system is finite, I would like the authors to comment or clarify this point if they want to make statements about the consequences of Lifshitz invariants.

We definitely agree that the gauge transformation that eliminates Lifshitz invariants cannot be performed in non-uniform systems, where magnetization emerging due to Lifshitz invariants makes the boundary condition non-trivial. Our statement referred to narrow and long wires $L \gg \xi_0$, where ξ_0 is the coherence length, where the absolute value of the order parameter can be considered constant within layers, and magnetic field is treated as uniform. We added this disclaimer into revised introduction. Note that we also checked experimentally that NRC does not depend on the wire length, Fig. 2b. This makes it less likely that the observed effect is due to finite size effects.

8. At the end of the main text part, there are several estimates of quantities, which depends on experimental parameters, but it is not clear how they were determined. To be more explicit, I would like to know how Δ in InAs (this is actually the proximity induced Δ , usually indicated as Δ^* - see e.g. Phys. Rev. Appl. 7, 034029 (2017), Kjaergaard et al.). Also, I would like to know how the mean free path l is determined. In the system there are actually two mean free paths, that for the epitaxial Al and that for InAs. But, experimentally, I would guess it is not obvious how to determine them separately.

The superconducting gap in InAs $\Delta_{InAs}^* \approx 200\mu eV$ has been measured in a similar heterostructure in [PRL 119, 136803 (2017)], while $\Delta_{Al}^* = 1.796K_B T_c \approx 230\mu eV$. For estimates we assume $\Delta_{InAs}^* \approx \Delta_{Al}^*$ which does not qualitatively affect our order-of-magnitude estimates of η . We have added references in the text. We used two mean free paths, $l_{Al}^m \approx 5$ nm and $l_{InAs}^m \approx 300$ nm. The latter is calculated using 2D gas mobility and density measures with the Al layer removed. We have 7nm Al of which we expect the top 2 nm to be oxidised. It is logical here to assume the mean free path in Al to be of the order of the thickness 5nm (see Phys. Rev. X 8, 031041 (2018)).

9. The authors write: ”Such mechanism was proposed to explain Josephson current non-reciprocity, in this case $\Delta I \propto \sin(2ql)$, where q is a field-dependent momentum shift and l is the length of the junction[24]. We studied length-dependence of ΔI for wires with $l = 2, 3$ and 5 μm , Fig. 2(b). The three curves are almost identical, ruling out Doppler shift as an origin of NRS in our devices.” The finite momentum q (producing a so called helical superconducting state) is one of the main mechanisms considered by most theoretical works to explain the SC diode effect in plain films (not Josephson junction). In lateral SNS Josephson junctions the mechanism is slightly different since there is no order parameter in the N region. The supercurrent can be seen as carried by Andreev bound states. Ref 24 uses a phenomenological approach to describe the diode effect in SNS junctions. In Ref 24 the NRS is due to the parameter δ , which is the difference in phase offset (ϕ_{i0}) between the fundamental and the second harmonic in the coupling terms (see Eq. 1-2 in Ref 24). At a microscopic level this corresponds to the difference in the anomalous phase shift between the fundamental and the second harmonic in the current phase relation. The association made in Ref 24 between δ and ql might be justified on the basis of Ref. 22. But the fact that the NRS depends on the $\sin(2ql)$ is derived in Ref.24 considering a Josephson junction. This allows authors of Ref. 24 to expand the free energy in coupling terms containing the gaps of the two leads. I do not see how to extend this derivation to plain films, therefore the statement ” $\Delta I \propto \sin(2ql)$ ” in plain films cannot be considered established in the literature. Thus, the authors should either demonstrate this statement or show other (than [24]) references demonstrating the statement for plain films.

We agree with the reviewer that Josephson junctions, studied in Ref [24], require $L < \xi$, which is a regime opposite to what is studied in our paper. In the revised manuscript we added the following text before we discuss the cubic terms:

The observed NRC does not depend on the wire length, Fig. 2b, which rules out formation of spurious loops due to the presence of wire/contact boundaries. It also rules out size-dependent NRC due to magnetic field-dependent Doppler shift of the Cooper pair momenta expected in short ($l < \xi$) junctions [23,31].

10. In the Supplement the authors write: "The critical current through the ring is determined by the condition that at large enough I_{ext} , one of the wires (or arms of the ring) turns normal. (Experiment indicates that the switching happens in Al, i.e., wire 1, see below.)"

And then later...

"The measurement shows also that the switching current does not differ much from its $B_{\text{perp}}=0$ value (see Fig.S2c), which indicates that the critical current is determined by wire 1, as we assumed in Eq. (S9)". This latter statement is supposed to demonstrate the former assumption, stated just before Eq. S9, that it is Al that switches first. What it is not clear to me is why the proximitized InAs should not switch normal first (the system would still be superconducting since superconducting Al would shunt it).

Per Referee's question, if the critical current of Al is larger than the external current then indeed Al can carry all the current without switching to a normal state. In the phase-locked case, $\phi_1 = \phi_2$, the distribution of externally applied supercurrent in the two wires is determined by the ratio of kinetic inductances, $I_2/I_1 = L_1/L_2 \equiv \eta$. This dimensionless parameter has to be compared to the ratio of critical currents $I_{2,c}/I_{1,c}$ to determine which wire reaches the critical current first. In our experiment we cannot measure these ratios independently, but expect both to be much smaller than 1. Therefore it is not obvious which wire would reach the critical current first. However, the slope $d\Delta I/dB_{\perp}$ of non-reciprocal critical current, Eq. (S11), will differ by a large factor $1/\eta$, depending on which wire turns normal first. Thus, the slope of non-reciprocity allows us to determine that Al (wire 1) turns normal first.

We have added Fig. 4c in the main text to illustrate the distribution of currents in the wires and the critical current. We also added a section "Determining total critical current in a 2 wire model" in the supplement.

11. List of minor remarks:

a) Many references have been published, yet they are still indicated with their preprint arxiv number. Please update the manuscript citing references in their final published form. b) Many Latex references are broken, both in the main text and in the Supp. Mat., so that reference to some Figures and Equations are missing (?? sign appears). This makes reading difficult. Please fix already at this stage. c) The symbol "l" is used both as label for the mean free path and for the total length in $\sin(2ql)$.

Many of the arxiv preprints were not published at the time of the submission, we updated references to the published work and will do it again prior to the final publication. We also cleared Latex references.

Reply to Reviewer 2

1) In the abstract and in the introduction it is stressed that in homogeneous superconductor Rashba contributions do not play a role for NRS. However, this is not true in general as it is also mentioned by the authors itself, for instance regarding so-called Josephson diodes (e.g. in the Nat. Nanotech. paper by Baudgardner et al. the dominant role is believed to be due to Rashba contributions). These considerations should be amended and clarified. Furthermore, it is stated that the presented model can explain even older observations, but it is not clear how- and where it is discussed in the paper this point. In addition, many recent works on the present topic are not mentioned. In summary, I definitely suggest a revision of both abstract and introduction in order to clarify both the main results of the present manuscript and to avoid misunderstanding/misconceptions to non expert readers.

We modified the abstract and introduction to clarify that Rashba term should not play a role in long $L \gg \xi_0$ nanowires. We now also mention in the introduction that diamagnetic currents may be responsible for the appearance of NRC in Ref. [1]. The list of relevant references is updated with references [13-28]. Nat Commun 13, 4266(2022), arXiv:2110.01067, Nat. Phys. 18, 1221-1227(2022), Nat Commun 12, 2703(2021), Appl. Phys. Lett. 121, 102601(2022), J. Exp. Theor. Phys. 135, 226-230(2022), arXiv:2205.09276, Nat. Nanotechnol. 17, 39-44(2022), Nat. Phys. 18, 1228-1233(2022), Nat. Mater. 21, 1008-1013(2022), arXiv:2206.08471, Nat Commun 13, 3658(2022), Nature 604, 653-656(2022), arXiv:2210.02644, Nano Letters 10.1021/acs.nanolett.2c02899(2022).

2) There is some confusion in the orientation of the magnetic field used in the various measurements and the symbols used in the text. Since this is a relevant point, this should be clarified with care.

We use B_{\parallel} and B_{\perp} for in-plane field parallel and perpendicular to the wire, and B_z for the out-of-plane field. We checked that these fields are properly introduced in the main text and shown in Fig. 1d.

3) In Fig.1 the asymmetric switching current contributions does not vanish for $B=0$, why?

There is an uncompensated voltage present in the measuring circuit, which originates from the presence of burden voltage on the amplifier input and some uncompensated voltages in the fridge wiring (these voltages generate an offset current across the 100 k Ω resistor used as a current source). This offset current is constant throughout each cooldown. We *assume* that $\Delta I = 0$ at $B = 0$ and most of the data was adjusted accordingly. In the revised manuscript all data, including Fig. 1, is adjusted accordingly and we added the following statement to the Methods:

The data has been corrected for an instrumental cooldown-dependent constant current offset (generated in the circuit by uncompensated voltages in the system and limited by a 100 k Ω current source resistor) to insure that $\Delta I = 0$ at $B = 0$.

4) Have the authors measured also the retrapping current? Does this quantity share analog behaviours?

We do not measure retrapping current due to excessive heating in the normal state.

5) It is mentioned that a small gate modulation is present by acting with a top gate on a region where Al has been removed and this influence supercurrent amplitude. The authors stated that this is consistent with previous measurements on superconducting film and the presence of quasiparticles. I was wondering if this can be also explained by a simpler Jo-fet like contributions, where the gate simply tune the electron density on a pocket/normal region?

In order to deplete electron carriers in InAs we apply large negative gate voltage. Negative gate voltage also depletes carriers in Al (albeit negligibly small fraction) and, thus, should result in the *decrease* of I_c , contrary to the observed increase. The observed increase of the switching current may result from the reduction of quantum fluctuations due to the reduction of InAs volume for Cooper pairs to enter and, as a consequence, increasing switching current to be closer to the value of the critical current. We added this discussion into the text.

6) It is stated that the symmetric and asymmetric contributions have different temperature dependence with different origins. This is also showed in the supplement, but it is not fully justified. Could the authors provide more quantitative analysis and comments regarding the different origin of the se behaviours?

There are three observations that lead us to the conclusion that field-dependent symmetric and asymmetric contributions to the critical current have different origins: (i) these contributions have different field dependence ($\cos(2\theta)$) vs $\sin(\theta)$, (ii) they have different energy scales ($0.3 T_c$ vs $> 0.6 T_c$), and (iii) symmetric contributions has been reported in previous works on homogeneous Al nanowires with negligibly small spin-orbit interactions, where no NRC has been observed.

7) It is often introduced an efficiency parameter in order to estimate rectification, e.g. $\eta = \delta I_{sw} / (2 \langle I_{sw} \rangle)$. Could the authors report a quantitative analysis of this quantity?

We added $\Delta I / \langle I_{sw} \rangle$ scale to the Fig. 1.

8) It is not clear to me the figure where different nanowire lengths are studied (to rule out Doppler shift like contributions). Indeed, the investigated configuration is not the same as the ones for JDE. I suggest at least to move this discussion on the supplement, or to remove it .

In the revised manuscript we have added the following paragraph in the main text:

The observed NRC does not depend on the wire length, Fig. 2b, which rules out formation of spurious loops due to the presence of wire/contact boundaries. It also rules out size-dependent NRC due to magnetic field-dependent Doppler shift of the Cooper pair momenta expected in short ($l < \xi$) junctions [24,33].

9) The reported measures are explained in terms of a two wire models, pointing at an extrinsic effect for the observed NRS as in a multiply connected geometry (like in asymmetric squid geometry). However I have found in the SM that the distance d is 10 nm while I cannot find this value in the fabrication methods (it seems to me that there is a factor of two or so);

There was a typo in the materials section where this information was stated, the typo has been corrected.

in the supplement it is also stated that the first nw that become normal is the Al nw, called 1-th. However Al is also responsible for proximity effect of the InAs nw: it is somehow counterintuitive that the first

superconducting to normal transition takes place in the Al nw; in addition, I'm not sure regarding the conclusions that a reader can infer from this point, and in particular how this can be related to previous multilayer superconductor experiments and NRS. Overall, clear conclusions are missing.

In the phase-locked case, $\phi_1 = \phi_2$, the distribution of externally applied supercurrent in the two wires is determined by the ratio of kinetic inductances, $I_2/I_1 = L_1/L_2 \equiv \eta$. This dimensionless parameter has to be compared to the ratio of critical currents $I_{2,c}/I_{1,c}$ to determine which wire reaches the critical current first. In our experiment we cannot measure these ratios independently, but expect both to be much smaller than 1. Therefore it is not obvious which wire would reach the critical current first. However, the slope $d\Delta I/dB_\perp$ of non-reciprocal critical current, Eq. (S11), will differ by a large factor $1/\eta$, depending on which wire turns normal first. Thus, the slope of non-reciprocity allows us to determine that Al (wire 1) turns normal first (at that point the whole structure is turned normal).

We have added Fig. 4c in the main text to illustrate the distribution of currents in the wires and the critical current. We also added a section "Determining total critical current in a 2 wire model" in the supplement. We added the following statement in the main text: *We argue that this extrinsic origin of NRC is generic to multilayer superconductors and may provide a compelling explanation to the NRC observed in Ref. 1.*

10) (minor comments) all figure labels should be enlarged (especially in the inset); there is some typos in the supplement, e.g. the definition of δI_{sw} is not consistent with the one in the main text (wrong sign).

We thank reviewer for noting this discrepancy and corrected notations in the revised manuscript.

Reply to Reviewer 3

1) I found these two sentences wrong: "Theoretically it has been understood that linear-in-momentum energy terms, such as Rashba spin-orbit interaction or, more generally, any asymmetry-allowed Lifshits invariants[2], do not contribute to the supercurrent," and "It has been already demonstrated in the literature [2] that in uniform superconductors Lifshits invariants can be eliminated by a gauge transformation from both the Ginzburg-Landau (GL) equation and the expression for the supercurrent. Thus, they cannot lead to non-reciprocal contributions to the critical current or a kinetic inductance"

It is Known that linear-in-momentum Rashba SO interactions in a BCS superconductor CAN lead to non-reciprocal contribution in the critical current (see for example ref [3] of the manuscript and ref 10.1103/PhysRevLett.128.177001 for a more complete and general analysis). The ref [2] the authors use to exclude linear-in-momentum Rashba SO interaction is only limited to the GL approximation that is valid only for temperature close to T_c , a temperature range very different from their experiment.

There is a distinction between the terms linear in the Cooper pair momentum in the Ginsburg-Landau functional, i.e. the Lifshitz invariants, (more generally such terms in the Cooperon propagator and in the thermodynamic potential of a superconductor with a singlet pairing), and the linear in momentum spin-dependent Rashba term in the electron spectrum in the presence of Zeeman interaction. The term linear in the Cooper pair momentum can be indeed eliminated by a Galilean transformation in a uniform singlet superconductor (the thickness is less than the coherence length), but the Rashba-Zeeman effect is much more subtle, and definitely exists. We included the corresponding citations. Furthermore, as we mention in the paper ourselves, the Dresselhaus spin-orbit terms in the electron spectrum (both cubic and linear) in the presence of the Zeeman coupling result in non-reciprocity. We developed a microscopic theory of non-reciprocity emerging in systems with cubic and linear Dresselhaus terms in the electron spectrum, details will be presented in a separate paper. Here, based on symmetry arguments we demonstrate that in the presence of Dresselhaus-related effects the non-reciprocity signal is strongly anisotropic, which is not observed in our experiments. Note that it was shown [G.W. Winkler et al., Phys. Rev. B 99, 245408 (2019)] that the Dresselhaus spin-orbit interaction is comparable or even stronger than the Rashba terms in heterostructures studied in our work. Thus, the absence of anisotropy of non-reciprocity provides an additional argument why the diamagnetic mechanism presented here can be an interesting and viable alternative to intrinsic, e.g., spin-orbit related mechanism.

We note that when effects of non-uniformity are present, the Lifshitz invariants nontrivially contribute to boundary conditions via magnetization, and cannot be eliminated. However, the spin-orbit and Zeeman terms may result in nonreciprocity in a uniform structure. But once again, for InAs/superconductor heterostructures the corresponding effect has to be anisotropic due to Dresselhaus-related contribution.

In the revised manuscript we clarified our statement in the introduction: *"It has been demonstrated in*

the literature [Agterberg 2012] that in uniform singlet superconductors in constant magnetic field the Lifshits invariants can be eliminated by a gauge (Galilean) transformation from both the Ginzburg-Landau (GL) equation and the expression for the supercurrent, so that **linear in Cooper pair momentum terms do not lead to non-reciprocity**" and we made clarifying corrections throughout the text.

2) Sign changes of the diode effect. It's not true that sign changes are not possible for diode effect based on Rashba SO interaction as mentioned for example in this sentence: "Non-monotonic field dependence and multiple sign changes of ΔI at high fields experimentally further exclude these mechanisms, since otherwise an intrinsic property of the material (vector c) would require to change sign as a function of B ." As shown in ref [3] of the manuscript and ref 10.1103/PhysRevLett.128.177001 a sign change is a peculiar characteristic of SO based diode effects.

The referee is correct in pointing out that both references predict a sign change of NRC, albeit with temperature dependence which is not seen in our experiments. In the revised version we removed the aforementioned statement. However, our main argument remains: the absence of crystallographic anisotropy of NRC rules out spin-orbit origin of the observed NRC.

3) Model based on the formation of vortices not very convincing. I found the model based on the interference between two superconductors not very convincing in few points:

3)a. Periodicity of the oscillations: in figure 1c I see two nodes in the ΔI , one at 200 mT and one at 400 mT, is this behavior confirmed in all the other samples? For example, in fig2 a (blue line) I see a node at -220mT and one at 300 mT and in fig S4 b the nodes are at -220 mT and 320 mT. If the origin of these nodes come from interference effects like squids the periodicity should be respected.

We revised the model description and provide a detailed qualitative discussion that includes a revised Fig.4.

We note that while there are similarities between NRC in a ring and in a double-layer superconductor there are some subtle differences. In a ring the area is well defined by the ring geometry, while the field at which a Josephson vortex is formed in a double-layer superconductor depends on interlayer Josephson coupling and slightly varies across the wafer. Thus, we do not expect the first node to be at exactly the same field in different samples. The second node also fluctuates, in some devices NRC is suppressed at fields smaller than twice the field of the first node. We added the following sentence to the main text:

"Josephson coupling \mathcal{E}_J is exponentially sensitive to the thickness of the InGaAs spacer and we expect slight variations of the period δB_{\perp} between the samples."

3)b. The model predicts supercurrent diode effect generated by one vortex of 0.5 μm size in the middle of the wire, while the generation of multiple vortices at large fields is excluded. How is it possible? What would be the impact of multiple vortices?

Generation of multiple vortices is not excluded, they will add or subtract identical diamagnetic currents at separate locations along the wires. Thus, appearance of multiple vortices will not change the critical current and critical current non-reciprocity. We added the following sentence to the main text:

"Generation of multiple Josephson vortices does not modify NRC compared to a single vortex case unless the vortices significantly overlap so that the maximum of $I_v(x)$ exceeds its single-vortex value. "

3)c. Knowing that also SO interaction linear-in-momentum can generate similar features (see point 2) in the same field range (if you consider the large g factor of InAs), how this hypothesis can be excluded? If confirmed this would make the experimental finding even more interesting.

In InAs QWs the strength of crystallography-dependent Dresselhaus SO is of the same order or exceeds [G.W.Winkler et al., Phys. Rev. B 99, 245408 (2019)] the crystallography-independent Rashba spin-orbit coupling. Thus, we expect spin-orbit-related effects to depend on the crystallographic orientation, which is not seen in reported experiments as emphasised in Fig. 2a. Currently we are performing a follow-up experiments aimed to detect the non-reciprocal corrections to the kinetic inductance, which originate from the Dresselhaus spin-orbit terms, Eq. 1. These measurements use rf techniques to boost the detection resolution by more than an order of magnitude.

4) Control device. In figure S6 the authors show the measurement on a control device of pure Al claiming that diode effect is not present. But, in panel b a sizable ΔI of -0.4 μA is visible and is larger than the

maximal ΔI observed in the devices 0.2 μA . How is it possible? In the control case there is no field dependence but the diode effect is larger or comparable with original devices.

There is an uncompensated voltage present in the measuring circuit, which originates from the presence of burden voltage on the amplifier input and some uncompensated voltages in the fridge wiring (these voltages generate an offset current across the 100 $\text{k}\Omega$ resistor used as a current source). This offset current is constant throughout each cooldown. We *assume* that $\Delta I = 0$ at $B = 0$ and most of the data was adjusted accordingly. In the revised manuscript all data is adjusted to have $\Delta I = 0$ at $B = 0$ and we added the following statement to the Methods:

The data has been corrected for an instrumental cooldown-dependent constant current offset (generated across a 100 $\text{k}\Omega$ current source resistor by uncompensated voltages in the system) to insure that $\Delta I = 0$ at $B = 0$.

REVIEWER COMMENTS

Reviewer #1 (Remarks to the Author):

The answers provided by the authors to my questions are mostly satisfactory. However, I still have some comments on the new version. I would like that an answer to these comments is given before publication.

1.

-The authors write (p.3)

"The term "superconducting diode effect" has been used to describe NRC in different systems, including thin superconducting films [14–20] and Josephson junctions [21–28]. In the former, the superconducting-to-normal transition is determined by the strength and symmetry of the flux pinning potential."

I disagree with the statement that in SC films the critical current (or the switch to the normal state) is (solely) attributed to vortices. Most of the theory and experiments discussing the diode effect in thin films with Rashba spin-orbit interaction investigate an intrinsic effect triggered only by the in-plane field. It is perhaps possible that the outcome of *some* experiments might have been influenced by vortex physics not adequately kept into account. However, the conclusion that all research field on the intrinsic diode effect in thin films can be just reduced to vortex physics is in my view absolutely not justified.

Some references:

Theory:

N. F. Q. Yuan and L. Fu, Supercurrent diode effect and finite-momentum superconductors, Proceedings of the National Academy of Sciences 119, e2119548119 (2022);

A. Daido, Y. Ikeda, and Y. Yanase, Intrinsic superconducting diode effect, Phys. Rev. Lett. 128, 037001 (2022);

J. J. He, Y. Tanaka, and N. Nagaosa, A phenomenological theory of superconductor diodes, New Journal of Physics 24, 053014 (2022);

S. Ilić and F. S. Bergeret, Theory of the Supercurrent Diode Effect in Rashba Superconductors with Arbitrary Disorder, Phys. Rev. Lett. 128, 177001 (2022).

Experiment:

Narita, H., Ishizuka, J., Kawarazaki, R. et al. Field-free superconducting diode effect in noncentrosymmetric superconductor/ferromagnet multilayers. Nat. Nanotechnol. 17, 823–828 (2022). <https://doi.org/10.1038/s41565-022-01159-4>

This latter reference is important: in the first version of the manuscript the authors wrote that that "the manuscript provides a compelling explanation to previously published results.". In my first question in the first round of review I asked the authors which published results the manuscript was providing a compelling explanation to. In the new version the authors have left only Ref 1 by the group of T. Ono (in the new version I read "Our findings show that the extrinsic contribution to NRC is generic to multilayer superconductors, and may provide a compelling explanation to the NRC observed in Ref. 1."). In my view, the aforementioned new paper by the group of T. Ono (Nat. Nanotechnol. 17, 823–828 (2022)) seems to exclude an explanation in terms of the arguments provided by the authors in the present manuscript (i.e., a diamagnetic mechanism). Therefore I still think (to the best of my knowledge) that there is no work in the literature to which the authors provide a compelling explanation to the NRC.

As a side remark, please correct the typo "includin".

2.

-The authors write (p.7)

"The observed NRC does not depend on the wire length, Fig. 2b, which rules out formation of spurious loops due to the presence of wire/contact boundaries. It also rules out size-dependent NRC due to magnetic field-dependent Doppler shift of the Cooper pair momenta expected in short ($l < \xi$) junctions [24, 33]."

This is an important sentence, which appears in the new version. It is cited in the answers to the other Referees as well.

First of all, Ref 24 is totally misplaced here (and, by the way, its arxiv number is missing). Ref. 33 also does not seem to me to fit very well here. Ref 33 is about a short SNS junction system, not a

wire. It is not clear to me how it would support the statement that "It also rules out size-dependent NRC due to magnetic field-dependent Doppler shift". The authors should explain it.

3.

I would finally like to comment on the important Question 3c (of the first round of review) by the Referee 3, and to the corresponding answer given by the authors.

The main argument of the authors is that, since the Dresselhaus spin-orbit coupling is expected to be large (the authors cite PRB,99 245408 to corroborate the statement), then the effect should be lattice-orientation-dependent. Since experimentally it is not, then the spin-orbit-based mechanisms must be ruled out.

I disagree with the strong statement that the Dresselhaus spin-orbit must definitely be larger or comparable to Rashba. The cited reference PRB,99 245408 is about *calculations* on InAs *nanowires* (and not, as the authors wrote in the reply, on quantum wells). However, a recent paper [C Baumgartner et al (2022) J. Phys.: Condens. Matter 34 154005 , follow up of Ref 21], working on a system nearly identical to the one studied by the authors (InAs quantum well with epitaxial Al on top), showed that the Dresselhaus component of the spin-orbit is indeed there, but it is much smaller than the Rashba.

One could believe or not the outcome of that experiment, but the opposite statement that the Dresselhaus component of the spin-orbit is necessarily small is for me unjustified (and therefore the conclusion about the inconsistency of the spin-orbit mechanism as explanation for the effect is equally not justified, in my view).

Reviewer #2 (Remarks to the Author):

I have read the revised version of the manuscript.

The authors made substantial changes and now the main message of the article is clear, therefore in my view, the paper is suitable for publication.

Reviewer #3 (Remarks to the Author):

The manuscript by Ananthesh Sundaresh and co-workers: "Diamagnetic mechanism of supercurrent non-reciprocity in multilayered superconductors" investigates the supercurrent diode effect in nanowires fabricated from InAs/Al heterostructures. A clear effect is visible and depends mainly on the in plane magnetic field applied orthogonal to the nanowire. Differing from similar experimental they exclude the role of Rashba and Dresselhouse SO interaction in the asymmetry of the critical current and they develop a model based on the interference between two dissimilar superconductors similarly to an asymmetric SQUID.

In their second submission the authors have corrected and improved the manuscript. Most of my concerns have been resolved and the text is more clear and suitable for publication once these minor corrections have been done:

1-Not clear difference respect to JJs. In the main text it is not clear the difference between a JJ and their system: a long nanowire made of a multilayer superconductor, not a JJ in the long regime. In particular, I found these sentences misleading: "In Josephson junctions, a finite size $L < \xi_0$, where ξ_0 is the superconducting coherence length, is important for consideration of NRC. Here we restrict our discussion of the origin of NRC to long ($L \gg \xi_0$) nanowires." And "It also rules out size dependent NRC due to magnetic field-dependent Doppler shift of the Cooper pair momenta expected in short ($L < \xi$) junctions [24, 33]." Ref 24 and 33 refer to JJs not superconducting nanowires.

2-Definition of rectification used in fig1 c not correct. The authors should use the definition of rectification suggested by referee #2 at point 7 ($\eta = \delta I_{sw}/(2\langle I_{sw} \rangle)$) which is the one commonly used and reaches 100% for ideal rectification. Now in the figure is reported $\delta I_{sw}/\langle I_{sw} \rangle$ that brings to the nonphysical result of 200% for ideal rectification.

RESPONSE TO REVIEWERS' COMMENTS

Reviewer 1 (Remarks to the Author):

The answers provided by the authors to my questions are mostly satisfactory. However, I still have some comments on the new version. I would like that an answer to these comments is given before publication.

1.-The authors write (p.3)

“The term “superconducting diode effect” has been used to describe NRC in different systems, including thin superconducting films [14–20] and Josephson junctions [21–28]. In the former, the superconducting-to-normal transition is determined by the strength and symmetry of the flux pinning potential.” I disagree with the statement that in SC films the critical current (or the switch to the normal state) is (solely) attributed to vortices. Most of the theory and experiments discussing the diode effect in thin films with Rashba spin-orbit interaction investigate an intrinsic effect triggered only by the in-plane field. It is perhaps possible that the outcome of *some* experiments might have been influenced by vortex physics not adequately kept into account. However, the conclusion that all research field on the intrinsic diode effect in thin films can be just reduced to vortex physics is in my view absolutely not justified.

An assertion of the Reviewer that “Most of the theory and experiments discussing the diode effect in thin films with Rashba spin-orbit interaction investigate an intrinsic effect triggered only by the in-plane field” misrepresents the experimental results. In ALL referenced papers NRC is observed ONLY when magnetic fluxes are present in the films, and no NRC is observed in the absence of fluxes (even in the presence of in-plane field).:

- In Ref [Bauriedl2022], it is clearly shown in Fig. 2f that there is no NRC without out-of-plane B_z component. The critical current decreases linearly with B_z (Fig. 2d), which means that critical current is limited by the flux depinning rather than by the Cooper-pair depairing.
- In Ref [Lin2022], in order to see NRC authors need to “train” the sample by first applying out-of-plane field, a conventional way to form and trap fluxes in thin films.
- Ref [Lyu2021] NDC in conventional SC with artificial flux pinning cites (requires the presence of vortices to observe NRC).
- Ref [Suri] “We observe a large diode efficiency of 30% when an out-of-plane magnetic field as small as 25mT is applied... we find that the diode effect vanishes when the magnetic field is parallel to the sample plane. Our observations are consistent with the critical current being determined by the vortex surface barrier.”
- Ref. [Ustavshikov2022] diode effect is non-zero only for $B_z \neq 0$, Fig. 2b, NRC requires formation of fluxes.
- Ref [Hou2022] diode effect in EuS/Nb is non-zero only for $B_z \neq 0$. Authors emphasize “we attribute the apparent in-plane diode effect in Fig. 1E to the residual out-of-plane component of the magnetic field.”

“Superconducting diode effect” refers to the non-reciprocity of the critical current I_c , the value of I_c has different physical origins in different settings. In nanowires studied in our manuscript, I_c is limited by Cooper-pair depairing velocity (Bardeen limit), while in thin films in the presence of magnetic fluxes I_c is reduced and is determined by the flux depinning. These are different physical mechanisms *independent of the role of spin-orbit interactions*. How in-plane field modifies the “the strength and symmetry of the flux pinning potential” is not a topic of our paper.

Furthermore, in the recent preprint [Hou2022] after careful examination of FM/SC, HM/SC/FM and SC/OX/FM heterostructures authors clearly state that “Based on these observations, we conclude that

neither Rashba SOC nor interfacial exchange with the FM are essential in the observed Ic nonreciprocity” and attribute the observed NRC to diamagnetic currents generated by fringing magnetic fields from FM layer. In experiments of Hou et al critical current is limited by Cooper pair de-pairing and suggested mechanism of non-reciprocity is related to the physics discussed in our paper.

In the revised manuscript the paragraph in question now reads: *The term “superconducting diode effect” has been used to describe NRC in different systems, including thin superconducting films [Bauriedl2022, Jiang2022, Lyu2021, Suri2022, Ustauschikov2022, Hou2022] and Josephson junctions [Diez-Merida2021, Baumgartner2022, Pal2022, Jeon2022, Gupta2022, Golod2022, Wu2022, John2022, Bianca2022]. In the former experiments the presence of out-of-plane magnetic field and formation of vortices is essential for the observation of NRC, in this case the critical current is determined by the strength and symmetry of the flux pinning potential. In the latter case the critical current in Josephson junctions is determined by the overlap of Andreev states. In this paper we restrict our discussion to the origin of NRC in long nanowires, where critical current is determined by the de-pairing velocity of Cooper pairs (the Bardeen limit [Bardeen1962]).*

Some references:

Theory:

N. F. Q. Yuan and L. Fu, Supercurrent diode effect and finite-momentum superconductors, Proceedings of the National Academy of Sciences 119, e2119548119 (2022);

A. Daido, Y. Ikeda, and Y. Yanase, Intrinsic superconducting diode effect, Phys. Rev. Lett. 128, 037001 (2022);

J. J. He, Y. Tanaka, and N. Nagaosa, A phenomenological theory of superconductor diodes, New Journal of Physics 24, 053014 (2022);

S. Ilić and F. S. Bergeret, Theory of the Supercurrent Diode Effect in Rashba Superconductors with Arbitrary Disorder, Phys. Rev. Lett. 128, 177001 (2022).

The last three papers were cited as references [3,4,9], for the paper by Liang Fu we cited his earlier paper on the subject [12] and updated the reference in the revised manuscript.

Experiment:

Narita, H., Ishizuka, J., Kawarazaki, R. et al. Field-free superconducting diode effect in noncentrosymmetric superconductor/ferromagnet multilayers. Nat. Nanotechnol. 17, 823–828 (2022). This latter reference is important: in the first version of the manuscript the authors wrote that that “the manuscript provides a compelling explanation to previously published results.”. In my first question in the first round of review I asked the authors which published results the manuscript was providing a compelling explanation to. In the new version the authors have left only Ref 1 by the group of T. Ono (in the new version I read “Our findings show that the extrinsic contribution to NRC is generic to multilayer superconductors, and may provide a compelling explanation to the NRC observed in Ref. 1.”). In my view, the aforementioned new paper by the group of T. Ono (Nat. Nanotechnol. 17, 823–828 (2022)) seems to exclude an explanation in terms of the arguments provided by the authors in the present manuscript (i.e., a diamagnetic mechanism). Therefore I still think (to the best of my knowledge) that there is no work in the literature to which the authors provide a compelling explanation to the NRC. As a side remark, please correct the typo “includin”.

Here we disagree with the referee, the paper by Narita et al is fully consistent with our model. In the new experiment the authors inserted Cr film between Nb and V which is exactly our two-wire model with a fixed flux between the two layers. This flux, provided by the Cr layer, generates opposite diamagnetic currents in Nb and V layers which should lead to non-reciprocity. There is no experimental evidence of exchange interactions playing a role in the aforementioned experiment. We included this reference in the revised manuscript: *“Our findings show that the extrinsic contribution to NRC is generic to multilayer superconductors, and may provide a compelling explanation to the NRC observed in Ref. [Ando2022] and [Narita2022], in the latter work magnetic flux produced by a Cr layer generates opposite diamagnetic currents in the adjacent Nb and V layers.*

2. -The authors write (p.7)

“The observed NRC does not depend on the wire length, Fig. 2b, which rules out formation of spurious loops due to the presence of wire/contact boundaries. It also rules out size-dependent NRC due to magnetic field-dependent Doppler shift of the Cooper pair momenta expected in short ($l < \xi$) junctions [24, 33].” This is an important sentence, which appears in the new version. It is cited in the answers to the other Referees as well. First of all, Ref 24 is totally misplaced here (and, by the way, its arxiv number is missing). Ref. 33 also does not seem to me to fit very well here. Ref 33 is about a short SNS junction system, not a wire. It

is not clear to me how it would support the statement that “It also rules out size-dependent NRC due to magnetic field-dependent Doppler shift”. The authors should explain it.

We see that the aforementioned phrase causes confusion and removed it from the revised version. We left only the first sentence *The observed NRC does not depend on the wire length, Fig. 2b, which rules out formation of spurious loops due to the presence of wire/contact boundaries.*

3. I would finally like to comment on the important Question 3c (of the first round of review) by the Referee 3, and to the corresponding answer given by the authors. The main argument of the authors is that, since the Dresselhaus spin-orbit coupling is expected to be large (the authors cite PRB,99 245408 to corroborate the statement), then the effect should be lattice-orientation-dependent. Since experimentally it is not, then the spin-orbit-based mechanisms must be ruled out. I disagree with the strong statement that the Dresselhaus spin-orbit must definitely be larger or comparable to Rashba. The cited reference PRB,99 245408 is about *calculations* on InAs *nanowires* (and not, as the authors wrote in the reply, on quantum wells). However, a recent paper [C Baumgartner et al (2022) J. Phys.: Condens. Matter 34 154005, follow up of Ref 21], working on a system nearly identical to the one studied by the authors (InAs quantum well with epitaxial Al on top), showed that the Dresselhaus component of the spin-orbit is indeed there, but it is much smaller than the Rashba. One could believe or not the outcome of that experiment, but the opposite statement that the Dresselhaus component of the spin-orbit is necessarily small is for me unjustified (and therefore the conclusion about the inconsistency of the spin-orbit mechanism as explanation for the effect is equally not justified, in my view).

There is a clear misunderstanding which we try to clarify in the revised version. In the paragraph in question we discuss the ratio of **cubic** Dresselhaus terms and cubic in Q terms which originate from the presence of the Rashba terms in the energy spectrum (as we discussed in the introduction linear in Q terms in the GL functional do not contribute to the NRC in uniform materials). Such a ratio was not measured experimentally as far as we know and we use theoretical values for the estimate. Spin-orbit anisotropy in InAs QWs, which includes linear Rashba and Dresselhaus terms, has been studied experimentally using optical and transport techniques, although we have reservations regarding data analysis and the actual numbers reported in these papers: in the analysis of optical data Giglberger et al. [PRB, 75 (2007)] assume that spin relaxation time is isotropic (unjustifiably), in the analysis of magnetoresistance Farzaneh et al [arXiv:2208.06050] do not account for electron-electron interactions (which are of the same order as WAL), and in the aforementioned paper anisotropy is found by modeling the system using KWANT which we found to produce NRC in long wires with only Rashba interactions, a clear violation of Galilean invariance of the system.

Nevertheless, we agree that spin-orbit anisotropy reported in previous works can serve as a lower bound to the expected spin-orbit-related NRC anisotropy. In the revised manuscript we modified the discussion of anisotropy as follows: *The cubic (and generally all odd) in Cooper pair momentum terms originating from the Rashba electronic interactions, when added with the Dresselhaus-induced terms, will produce anisotropy in the absolute value of NRC, and, in particular, different values of non-reciprocal asymmetrical component of the current for those two directions. Theoretical investigation of electronic spectra of these systems [Winkler2019] suggests that in narrow InAs quantum wells cubic Dresselhaus terms are larger than the Rashba terms. The lower limit for the value of the Dresselhaus contribution can be extracted from the total spin-orbit anisotropy (which is defined by the ratio between a linear Rashba, and a linear and cubic Dresselhaus terms in electronic spectrum), which was measured to be 70% in spin-galvanic and circular photogalvanic experiments [Giglberger2007] and > 15% in transport experiments [Farzaneh2022,Baumgartner2022a]. Such anisotropies must result in the corresponding crystallographic anisotropy of the NRC, which is not observed in our experiments, Fig. 2(a).*

Reviewer 2 (Remarks to the Author):

I have read the revised version of the manuscript. The authors made substantial changes and now the main message of the article is clear, therefore in my view, the paper is suitable for publication.

Reviewer 3 (Remarks to the Author):

In their second submission the authors have corrected and improved the manuscript. Most of my concerns have been resolved and the text is more clear and suitable for publication once these minor corrections have been done:

1-Not clear difference respect to JJs. In the main text it is not clear the difference between a JJ and their system: a long nanowire made of a multilayer superconductor, not a JJ in the long regime. In particular, I found these sentences misleading: “In Josephson junctions, a finite size $L < \xi_0$, where ξ_0 is the superconducting coherence length, is important for consideration of NRC. Here we restrict our discussion of the origin of NRC to long ($L \gg \xi_0$) nanowires.” And “It also rules out size dependent NRC due to magnetic field-dependent Doppler shift of the Cooper pair momenta expected in short ($l < \xi$) junctions [24, 33].” Ref 24 and 33 refer to JJs not superconducting nanowires.

We see that the aforementioned phrase causes confusion and removed it from the revised version. We left only the first sentence *The observed NRC does not depend on the wire length, Fig. 2b, which rules out formation of spurious loops due to the presence of wire/contact boundaries.*

2-Definition of rectification used in fig1 c not correct. The authors should use the definition of rectification suggested by referee 2 at point 7 ($\eta = \delta I_{sw}/(2\langle I_{sw} \rangle)$) which is the one commonly used and reaches 100% for ideal rectification. Now in the figure is reported $\delta I_{sw}/\langle I_{sw} \rangle$ that brings to the nonphysical result of 200% for ideal rectification.

We agree that $\delta I_{sw}/(2\langle I_{sw} \rangle)$ is a more appropriate definition and divided the scale in Fig. 1c by a factor of 2 as suggested.

REVIEWERS' COMMENTS

Reviewer #1 (Remarks to the Author):

I find the authors' responses satisfactory, so I recommend publication of the manuscript.